# CaStRL: Context-Aware State Representation Learning with Transformer

## Abstract

Learning a versatile representation from high-dimensional observation data is a crucial stepping stone for building autonomous agents capable of effective decision-making in various downstream tasks. Yet, learning such a representation without additional supervisory signals poses formidable practical challenges. In this work, we introduce Context-Aware State Representation Learning (CaStRL), a novel unsupervised representation pretraining approach that combines the strength of generative autoregressive modeling with the pretraining-finetuning paradigm. To encourage CaStRL to grasp the underlying dynamics information of the environment, we enforce it to jointly learn the latent state representation along with the contexts that influence the model's ability to learn a generalizable representation for control tasks. In CaStRL, we first employ the Video-Swin Transformer as a vision encoder, customizing it to support auto-regressive modeling through the incorporation of a causal attention mask. Then, we design Context-GPT to learn context from historical sequences of state representation, which drives the model towards capturing global structural patterns by propagating information across extended time horizons. This significantly improves the adaptability of the learned representation for diverse control tasks. By emphasizing reward-free evaluation and limited data constraints in both pretraining and fine-tuning stages, we find, across a wide range of Atari experiments, that pretrained representations can substantially facilitate downstream learning efficiency.

## 1 Introduction

State representations are often crucial for effective Reinforcement Learning (RL), activity recognition, and other downstream tasks (van den Oord et al., 2018; Han et al., 2019; Chen et al., 2022). A representation that focuses narrowly on relevant information can facilitate learning an effective control policy, while a representation that includes irrelevant noise can be detrimental. Using handcrafted features or high-dimensional raw sensory data to define the state are two simple approaches. Unfortunately, each has significant disadvantages. Handcrafting the state for Atari environments (Mnih et al., 2015), for example, requires certain domain expertise and additional human oversight. Often, manually designed state representations suffer certain drawbacks such as presence of irrelevant information, provide insufficient coverage of complex dependencies within state space. In contrast, raw observation data is typically high-dimensional, and includes irrelevant information that makes it difficult to leverage any synergistic benefits from multiple down-stream tasks.

Deep Reinforcement Learning (DRL) has gained extensive attraction in recent years, especially, its ability to compress high-dimensional observation into a compact representation (Silver et al., 2016; Berner et al., 2019). But,its extensive usage prevented from sample efficiency problem: requires large amount of data which leads to lengthy training time (Dulac-Arnold et al., 2019). To address this issue, the focus has been shifted to incorporating inductive bias into DRL framework such as exploring parameterized neural network architecture (Zambaldi et al., 2018). Choice of DRL architecture has been explored but not limited to state representation learning objective. Yet, what type of DRL architecture is suitable for learning generalizable state representation remains elusive. The rise of the Transformer architecture (Vaswani et al., 2017) has revolutionized the learning paradigm in various domain. For instance, The profound success achieved by BERT (Devlin et al., 2019) and GPT model (Radford et al., 2018) in machine translation, language understanding, has sparked a big frenzy within machine learning community. As a result, this led to widespread adoption of

the pretrain-then-finetune paradigm (Howard & Ruder, 2018) as popular method to progressively enhance performance across wide-range of downstream learning tasks including imitation learning (Schwarzer et al., 2021). The recent works have shown that transformer architecture enjoys great extent of scalability and suitable for modeling long decencies such sequential decision making (Khan et al., 2021). Followed by such success, there has been increasing level of interest in adopting transformer in RL, in particular, top choice for learning state representation.

We consider the question: how can learned state representations be made more generalizable for downstream learning tasks? To address this question, we introduce Context-Aware State Representation Learning (CaStRL), a novel unsupervised representation pretraining approach. CaStRL not only emphasizes state representation learning, but also integrates context-awareness, where **context** pertains to features characterized by global structures that enable the propagation of information across extended time horizons. By including contextual information as regularization, our approach enhances the quality and relevance of learned state representations, making them more suitable for downstream control tasks. In our proposed approach, firstly, we employ the Video-Swin Transformer (Liu et al., 2021) as a vision encoder, tailoring it to support auto-regressive modeling through the incorporation of a causal-attention mask. Then, we design Context-GPT 4.3 aims to learn underlying latent state representation with its context. Whereas, **contexts** encapsulate information pertaining to all preceding states to enable jointly learning state-representation along with its context. Our empirical results demonstrate significant performance gains with CaStRL, showcasing the effectiveness of our unsupervised pretraining framework in enabling the model to rapidly emulate behaviors across multiple environments simultaneously.

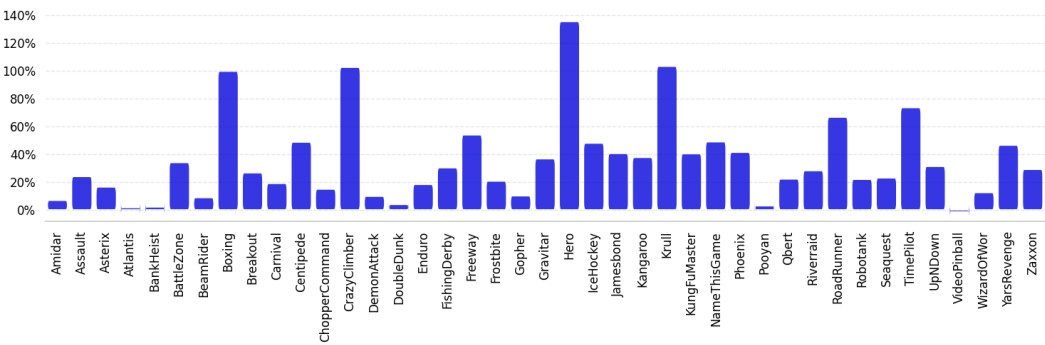

Figure 1: Performance of the top 3 CaStRL rollouts, normalized by the best training dataset scores.

The CaStRL framework has demonstrated significant performance gains and stability in multi-game settings. We evaluate CaStRL's capability to surpass the best training demonstrations and measure its deviation from random behavior. Figure 1 presents the normalized scores using the best DQN scores observed in the training dataset, following the Top-3 Metric as described in (Lee et al., 2022).

Our contributions are threefold; First, we propose CaStRL, a novel unsupervised representation pretraining approach for jointly learning latent state representation and its context, that influence the model's ability to learn a versatile representation for control tasks. Second, we design Context-GPT to learn context from historical sequences of state representation, which drives the model towards capturing global structural patterns by propagating information across extended time horizons. This significantly improves the adaptability of the learned representation for diverse control tasks. Third, we conduct extensive experiments on a wide spectrum of 46 Atari 2600 video games (Mnih et al., 2015). By evaluating using 41 games for training and 5 games for evaluation, we assess the generalizability of the learned representation. By emphasizing reward-free evaluation and limited data constraints in both pretraining and fine-tuning stages, we find, across a wide range of Atari experiments, that pretrained representations can substantially facilitate downstream learning efficiency.

## 2 RELATED WORKS

Recent work has extensively explored learning representations directly from raw sensory data with deep neural networks (Lange et al., 2012; Böhmer et al., 2015; Wahlström et al., 2015). Using

DRL to learn latent states with minimal supervision or through fully unsupervised approaches has gained significant traction (Radford et al., 2015; Anand et al., 2019). The main intuition is not to mandate a model that is directly supervised using ground-truth states since underlying state is inaccessible. Instead, it leverages sequences of observations and actions from offline experience datasets, intensively benefiting from the power of autoregressive models like GPT (Radford et al., 2018) to learn generalizable representations for diverse control tasks.

**Reinforcement Learning as Sequence Modeling:** In light of the significant success achieved by Generative Pre-training Transformer (GPT) and Large Language Models (LLMs) (Brown et al., 2020) across a wide range of tasks, recent research, as seen in (Chen et al., 2021) and (Shang et al., 2021), has extended the concept of generative pretraining to the realm of offline reinforcement learning. This extension involves formulating the RL problem as a sequence modeling challenge, enabling the learning of representations conditioned on rewards or return-to-go (RTG).

**Pretraining For Reinforcement Learning:** Subsequently, (Lee et al., 2022) presented empirical evidence demonstrating the capability of Decision Transformer (DT) (Chen et al., 2021) in expert action inference, showing a notable transfer learning performance gain by using multi-game settings for pretraining. Other recent works, such as (Sun et al., 2023), introduce a self-supervised supervised learning approach that incorporates momentum encoders and control transformers as a pretraining framework. However, there remains a scarcity of works dedicated to the comprehensive exploration of state representation learning from sequences of visual observations.

**Vision Encoders For Control Tasks:** Encoding visual observations is a fundamental building block for various sequence modeling problems. In recent works focused on control tasks (Chen et al., 2021; Shang et al., 2021; Sun et al., 2023), the visual encoder has typically been either a Convolutional Neural Network (CNN) (LeCun et al., 1998) or a Vision Transformer (ViT) (Dosovitskiy et al., 2020). However, these encoders lack the ability to incorporate temporal information effectively resulting in a loss of the finer temporal details present in a sequence of observations within their high-dimensional representation spaces. To address this concern, one potential approach is to use a vision transformer model explicitly designed to account for temporal dimensions, like the Video Swin Transformer (Liu et al., 2021). While the Video-Swin Transformer has demonstrated remarkable success in learning representations from video clips, its design initially lacks a direct emphasis on addressing control tasks formulated as autoregressive models. This is primarily due to the inherent nature of its underlying attention mechanism — Shifted Window Masked Self-Attention (SW-MSA), which lacks the causality required for autoregressive modeling with Transformers. Another challenge hindering the Video-Swin Transformer's applicability in control tasks is its lower inductive bias compared to basic convolution encoders, resulting in slower training.

## 3 PRELIMINARIES

The main objective of this work is to learn a compact, generalizable state representation to solve downstream tasks where control tasks and environments are defined by Partially Observed Markov Decision Processes (POMDPs), since the underlying state of the environment is not directly accessible from available high-dimensional sensory observations. A POMDP is described by a tuple $\langle \mathcal{S}, \mathcal{A}, \mathcal{O}, \mathcal{T}, \mathcal{R}, \gamma, E \rangle$, where: $\mathcal{S}$ is state space; $\mathcal{A}$ is action space; $\mathcal{O}$ is observation space, denoting image collections rendered from the environment; $\mathcal{T} : \mathcal{S} \times \mathcal{A} \rightarrow \mathcal{S}$ is state transition dynamics, which specify the probability that action $a_t \in \mathcal{A}$ in state $s \in \mathcal{S}$ will lead to next state $s_{t+1}$; $\mathcal{R} : \mathcal{S} \times \mathcal{A} \rightarrow \mathbb{R}$ is the reward function; and $E$ is observation probability $E(o|s)$ of perceiving observation $o$ given state $s$. At each timestep t, the RL agent observes $o_t$ based on underlying state $s_t \in \mathcal{S}$, takes action $a_t \in \mathcal{A}$, and receives reward $r_t$ then transition to next state $s_{t+1}$. Given sequences of observation-action tuples of length $L$ in the replay buffer, $b_t = (o_{t-L}, a_{t-L}, o_{t-L+1}, a_{t-L+1}, \ldots, o_t)$, the agent takes action $a_t$ based on policy $\pi : a_t = \pi(b_t)$, the agent aims to learn optimal policy $\pi^*$ that maximizes the cumulative reward $\mathbb{E}_P[\sum_{t=1}^{\infty} \gamma^t r_t]$ (Sutton & Barto, 2005).

### 3.1 GOOD CRITERIA FOR STATE REPRESENTATION LEARNING

The main objective in State Representation Learning (SRL) is to learn an informative and sufficient abstract representation from high-dimensional raw sensory data. We review important facets of what defines a good state representation that facilitate downstream policy learning. From a broader

perspective, the state representation should retain sufficient, compact and generic information from the observation in order to solve the RL task and filter out redundant information. Evaluating the quality of learned state representation, we consider these dimension of state space:

- **Compactness:** a state representation should be low-dimensional to directly guide the RL agent on decision making processes. Such compact representation should possess essential and meaningful representation, while ignoring redundant features.

- **Sufficiency:** encoding high-dimensional observation into low-dimensional abstract representation must contain essential and sufficient information to solve downstream RL tasks. State representations that contain insufficient information may lead to sub-optimal policies.

- **Generalizability:** learning more generelizable representation across semantically similar environments, which include but not limited to new environment states or totally different environments, has been considered as a key characteristic to assess its effectiveness in downstream learning tasks.

Tasks with long time horizon and partial observability are key challenges to representation learning.

# 4 METHOD

CaStRL jointly learn the latent state representation along with the contexts that influence the model's ability to extract a generalizable representation for control tasks. To learn such representation, firstly, we employ the Video-Swin Transformer as our vision encoder, tailoring it to support auto-regressive modeling through the incorporation of a causal-attention mask. In the second part of our work, we design Context-GPT learns contexts given a sequence of state-representation aims to learn underlying latent state representation with its context, where contexts encapsulate information pertaining to all preceding states to enable jointly learning state-representation along with its context.

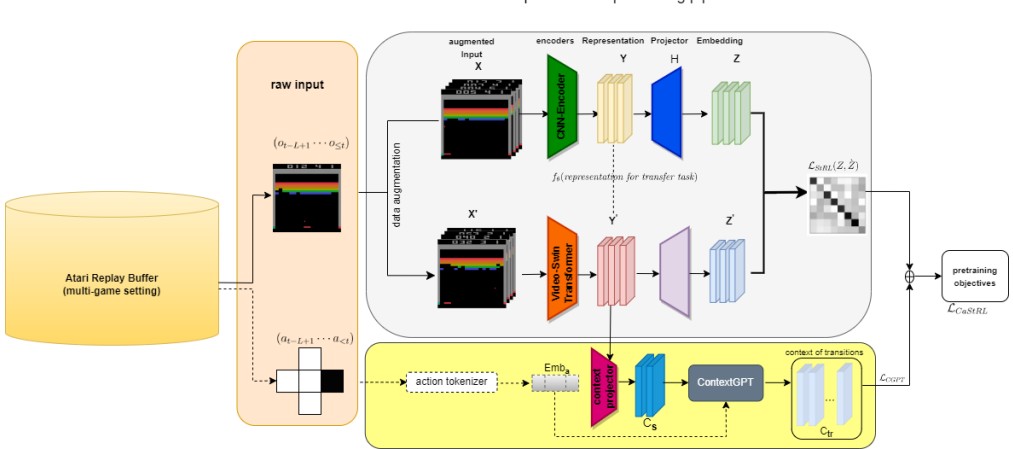

Figure 2: Illustration of the CaStRL pretraining pipeline.

## 4.1 VISION ENCODER

In this work, we employ Video Swin Transformer, originally proposed for video understanding tasks (Liu et al., 2021). We have adapted the Video-Swin Transformer by replacing the Shifted Window Multi-Head-Self-Attention (SW-MSA) with the Causal Shifted-Window Multi-Head-Self-Attention (CSW-MSA) (see details in appendix A.1). This tailored attention mechanism better suits the sequence modeling task, particularly in its autoregressive nature. Our primary focus is on leveraging Video-Swin Transformer as a foundational vision encoder capable of effectively learning both spatial and temporal representations. To the best of our knowledge, we are the first to include Causal-Attention into the Video-Swin architecture for state representation learning.

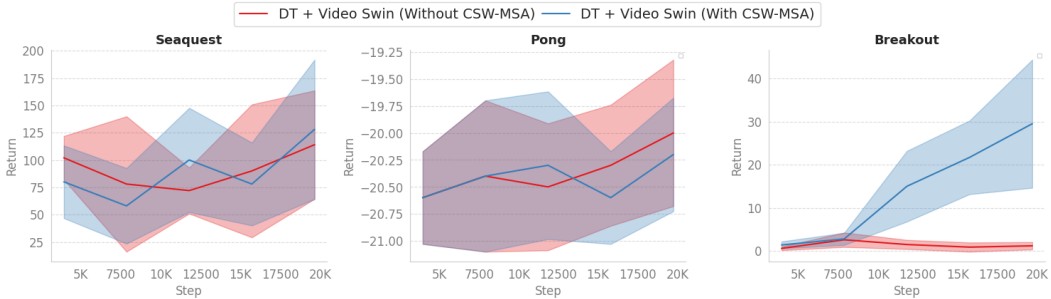

Figure 3: Performance of DT + Video Swin Transformer (With/Without CSW-MSA).

A fundamental question arises: to what extent does CSW-MSA broaden the scope of applicability of the Video-Swin Transformer in autoregressive modeling-based learning? To address this question, we replaced the convolutional encoder in the DT architecture (Chen et al., 2021) with the Video-Swin Transformer. We then evaluated the model's performance in Behavioral cloning (BC) (Pomerleau, 1988) tasks across some Atari environments in **single-game settings**, results shown in Figure 3. Another notable aspect we addressed is the lack of inductive bias in Video-Swin. During training, we simultaneously distilled knowledge from a convolution-based visual encoder (i.e., ResNet34 (He et al., 2015)) into our transformer-based encoder (Video-Swin). This resulted in the learned features from convolution encoders guiding the transformer encoder in the early stages of training. The knowledge distillation was achieved implicitly (Wang & Yoon, 2021), without the need for any additional modeling objectives. Further details on how this implicit knowledge distillation was achieved are discussed in section 4.2.

## 4.2 STATE REPRESENTATION LEARNING

The full architecture of our proposed Context-Aware State Representation Learning (CaStRL) approach is shown in Fig 2. CaStRL contains two learning components: state representation learning and transitions context learning. This section primarily addresses key questions related to the state representation learning component.

**Choice of Representation Learning Method:** While the Momentum encoder (He et al., 2019) has shown effectiveness as a powerful representation learning method (Sun et al., 2023), it does come with significant limitations regarding modeling flexibility. In this work, we emphasize the critical role of flexibility in the learning process. However, as the learning becomes more flexible, ensuring the effectiveness of what is learned becomes increasingly challenging. Issues such as degenerated solutions and overfitting come into play, leading most representation learning methods to incorporate regularization techniques. For instance, recent work (van den Oord et al., 2018) places higher requirements on the network to reduce the risk of overfitting, making it an effective regularization method for learning high-quality representations, as recognized in various studies.

Considering the modeling requirements outlined in section 4.1 for the visual encoder, we employ Variance-Invariance-Covariance Regularization (VICReg) (Bardes et al., 2021) as our chosen representation learning objective. Apart from VICReg's inherent flexibility, we chose it as the primary representation learning objective because it introduces two vision encoders that can be completely asymmetric with no shared structure or parameters. This property empowers our approach and contributes to its effectiveness in the task of representation learning. To achieve our objective of aligning Video Swin Transformer learned features with those of the CNN encoder during the pre-training phase. We employ both a CNN encoder (i.e., ResNet-like architecture) and a transformer encoder (i.e., Video-Swin) in our approach. Through minimizing the discrepancy between the representations acquired from these encoders, we enable Video-Swin to learn features comparable to those of the CNN encoder, resembling the knowledge distillation process between a teacher and a student network (Fan et al., 2018). The CNN encoder's inherent locality in its operations provides a substantial inductive bias, which the transformer lacks. Consequently, asserting that the CNN encoder learns features as poorly as the transformer encoder becomes a challenging proposition.

The VICReg objective effectively avoids the representation collapse problem. The steps of incorporating the VICReg objective into CaStRL are describe in Appendix 2.

## 4.3 Context-Aware Learning

In contrast to GPT (Brown et al., 2020) and LLMs, which excel at processing sequences of tokens (typically words) and harness the power of contexts to enable in-context learning (Dong et al., 2022), deriving context directly from a sequence of visual observations presents significantly greater challenges. To address this, we start by optimizing state representations, then use these optimized representations to learn context. This is a process of state representation optimization, followed by representations refinement guided by a Context-GPT objective. As shown in Figure 2, The learned state-representation $Y'$ is fed into a projection layer, which aims to summarize the state-representation in the feature dimension. This summarized output is referred to as the state-context representation or intermediate-context representation $C_s$. Next, a sequence of intermediate-context representations from previous steps are utilized as input for an autoregressive model - ContextGPT (CGPT). The autoregressive model summarizes the sequence in the temporal dimension, resulting in the transitions context $C_{tr} = cgpt(C_{s \leq t})$. During this process, each transition context at timestep $(k)$ is optimized to capture key information about all previous state-representations. The proper context learning approach ensures that the resulting transition context enables predictions in the future or past to be effectively independent of other timesteps, satisfying the Markovian property. As a result, the transitions context serves as a valuable representation, retaining the essential temporal dependencies.

In practice, there are various types of contexts that can induce context awareness. In this work, we explore two primary approaches for learning context:

1. Predicting multiple state representations in the future, akin to the way CPC (van den Oord et al., 2018) learns context. However, we do not employ contrastive losses in our method.

2. Addressing the Blank-filling problem (Raffel et al., 2020), which involves masking some of the tokens and then attempting to retrieve them. This task encourages the model to consider neighboring unmasked tokens to gather information about the missing tokens.

---

**Algorithm 1** Context GPT Loss (For Context-Aware StRL: **Context Type** ← **Next State**)

    ▷ **T**: Sequence Length
    ▷ **W**: Number of Future State Predictions to Make
    ▷ $\mathbf{s}^{[1:T]}$: Sequence of States — $s^1, s^2, \ldots, s^T$
    ▷ **Squeezer**: Squeezer Network for Intermediate Context Representations
    ▷ **ContextGPT**: GPT2-like Transformer
    ▷ **ContextDecoders**$^{[k]}$: $k^{th}$ Decoder Network
1: **procedure** CONTEXTGPTLOSS($s^{[1:T]}$)
      % Project state into intermediate context representations.
2:    $\mathbf{c_s}^{[1:T]} \leftarrow \mathbf{Squeezer}(s^{[1:T]})$
      % Predict the transitions context from a sequence of intermediate context representations.
3:    $\mathbf{c_{tr}}^{[1:T]} \leftarrow \mathbf{ContextGPT}(c_s^{[1:T]})$
      % Initialize Context-GPT loss.
4:    $\mathbf{loss} \leftarrow 0$
      % Computations of Context-GPT loss across shifted windows.
5:    **for** $t \leftarrow W + 1$ to $T$ **do**
6:        **for** $k \leftarrow 1$ to $W$ **do**
           % Relevance of the context at $(t - W)$ in predicting the state at $(t - W + k)$.
7:          $\rho_{\mathbf{k}} = \frac{W - k + 1}{W}$
           % Predict the state at $(t - W + k)$ using the context at $(t - W)$.
8:          $\hat{s}^{[t-W+k]} \leftarrow \mathbf{ContextDecoders}^{[k]}(c_{tr}^{[t-W]})$
           % Accumulate the discrepancy measure between $s^{[t-W+k]}$ and $\hat{s}^{[t-W+k]}$.
9:          $\mathbf{loss} \leftarrow \mathbf{loss} + \rho_k * \mathbf{VICReg}(s^{[t-W+k]}, \hat{s}^{[t-W+k]})$
10:        **end for**
11:    **end for**
12: **end procedure**

---

Using these two approaches, we introduce 3 variants of the Context-GPT objective:

1. **Context Type** ← **Next State**: This variant is tailored to support the learning of features extending over a longer time horizon, as elaborated in pseudocode 1.

2. **Context Type** $\leftarrow$ **Masked State**: This variant emphasizes context learning, with a focus on local context from neighboring states.

3. **Context Type** $\leftarrow$ **Masked Action**: While this variant involves supervised learning, it remains task-agnostic, as the model is not optimized to produce any specific BC policies.

The other variants of the Context-GPT objective are described in pseudocodes 3 and 4.

# 5 EXPERIMENTS

In our experiments, we focus our attention on using behavior cloning (BC) (Pomerleau, 1988) in multi-environment scenarios as a downstream task to assess the effectiveness of CaStRL. Instead of using setups that depend on rewards (Chen et al., 2021; Shang et al., 2021), we use BC to see how useful our learned state representation is independently of rewards. Our experiments focus on: examining the extent to which pretraining influences our model's performance; in-depth investigation into how incorporating or excluding specific context information influences our model's performance; and assessment of the performance of existing methods (Chen et al., 2021; Shang et al., 2021) in comparison to our framework.

## 5.1 EXPERIMENTAL SETUP

Throughout both the pretraining and finetuning phases, training takes several days on 4 V100 32GB GPUs. To manage this computational demand, our studies are based on a limited portion of the Atari offline experience datasets (Agarwal et al., 2019). In both the pretraining and fine-tuning phases, our experiments were conducted with limited data constraints. This decision serves a twofold purpose: to reduce training time and to gain insights into how our approach performs in situations with limited data availability.

**Pretraining and Finetuning Dataset:** For both the pretraining, finetuning phases, and even the baseline training, we used an existing offline experience datasets (Agarwal et al., 2019), and omitted the consideration of incorporating rewards into our training data, as the primary objective is to evaluate the CaStRL model in reward-free settings. This dataset consists of trajectories collected during the training of a DQN agent.

**Atari Environment Selection:** In alignment with the environment selection strategy outlined in Multi-Game DT (Lee et al., 2022), we used 41 environments in the pre-training phase while reserving 5 environments to assess CaStRL's generalizability. This encompasses a total of 46 Atari environments in our study.

**Dataset Size for Multi-game Experiments:** In all multi-game experiments, a limited subset of offline experience datasets was employed, with each environment accumulating $100K$ steps, totaling 4.1 million environment steps across the 41 environments. This is notably smaller than the training set of (Lee et al., 2022), where the same $41$ environments were used with 160 billion environment steps.

**Data Augmentation:** For data augmentation, we treat spatiotemporal aspects separately. Specifically, we applied color and noise-based augmentation individually to each frame while maintaining consistent geometric transformations across frames for each segment of the observation sequence. This strategy was implemented to prevent the model from excessively relying on low-level optical-flow features for learning, aligning with the approach mentioned in (Han et al., 2019).

**Generalization Evaluation:** To assess CaStRL's generalization capabilities, a multi-game experiments conducted based on the pretrained model. Specifically, we evaluate the model's performance on 5 held-out environments.

**Pretraining Procedure:** We use an embedding size of 96 for the Video Swin Transformer (Liu et al., 2021), complemented by our custom attention mechanism, CSW-MSA. In our architecture, we incorporate ResNet34 with minor adjustments (He et al., 2015). To facilitate tokenization in the temporal dimension, we introduce an embedding block consisting of a single 3D Convolution layer, followed by Batch Norm, and ReLU activation function. For VICReg, we use an expander network comprising 3 layers, each featuring a linear layer with a dimension of 1024, followed by a GELU activation function. The hyperparameters for VICReg, including $\gamma$, $\beta$, and $\eta$, are set to

1.0, 0.1, and 1.0, respectively. In the GPT module, both during pretraining and finetuning stage, we employ 6 layers and 4 attention heads. We set a context length of $T = 16$ and a context dimension of $dim_{context} = 192$. During the pretraining phase, we simultaneously optimize two objectives: VICReg and Context-GPT Loss. We conducted the pretraining for a total of $50$ epochs and put an explicit limit on the number of batch samples to $13.5K$ per epoch. This limitation was added to speed up the pretraining process. while CaStRL continued to perform well under these training constraints.

**Finetuning Procedure:** In the finetuning phase, we further optimize the pretrained model on the BC task for 10 epochs in a multi-game setting. During this process, we employ 41 environments to assess the effectiveness of CaStRL. Additionally, for the purpose of generalization testing, we finetune the unsupervised pretrained version on 5 environments that were not encountered during the pretraining phase. Similar to the pretraining phase, we impose an explicit limit on the number of batch samples, set to $13.5K$ per epoch. It could be problematic if the model forgets what it learned during the pre-training phase. This issue may arise due to a distribution shift, which is a result of not using augmentation during the finetuning phase and incorporating environments not seen during the pretraining phase for generalization testing. To overcome the challenge of catastrophic forgetting, we adopt a simple yet effective approach: we initially freeze all layers except the last Linear layer for the first epoch and subsequently unfreeze all layers, mirroring the approach proposed in (Howard & Ruder, 2018). We finetuned the entire model, as freezing the pretrained model has shown to be ineffective in handling complex environments (Sun et al., 2023). This decision is justified by the realization that what is learned during the pretraining phase may not always be directly exploitable during the finetuning phase, and some adaptation or refinement would be required.

**Baselines:** For DT (Chen et al., 2021) and Starformer (Shang et al., 2021) baselines, we replicate our finetuning experiments settings. Since training multi-game environments takes a prohibitively long time, we train DT and Starformer from scratch based on just one seed for evaluation. Nevertheless, we carefully replicate our finetuning experimental settings, ensuring an identical subset of the data was used. Additionally, to ensure a fair comparison with CaStRL, we matched the GPT model scale.

**Evaluation Metrics:** We assess the performance on individual Atari games relative to an assumed lower bound: optimizing a randomly initialized version of our model, i.e., without any pretraining. We then calculate the relative score of each game in reference to this baseline. See Appendix C.1 for more results.

## 5.2 EXPERIMENTAL RESULTS

We first evaluate how altering the context being learned affects BC task performance. To achieve this, we pretrained 7 different versions of our model, each targeting a distinct context type, and the results are presented in Figure 4. The best-performing context version is then compared against the baseline models in the BC task. The best-performing variant of CaStRL is then compared against the baseline models in the BC task, with the results shown in Figure 5.

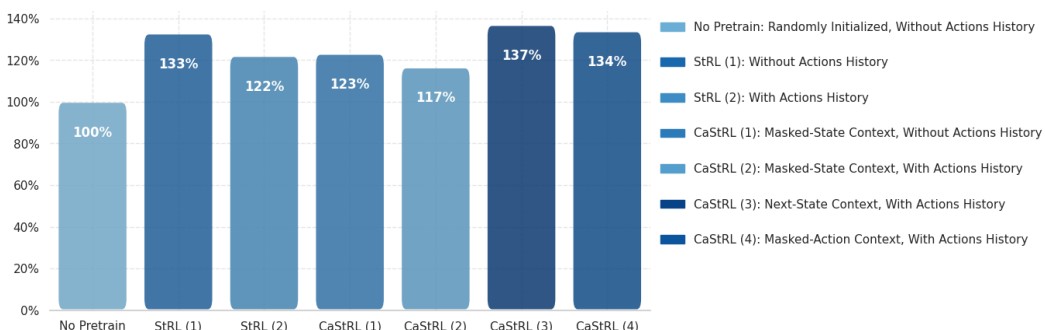

Figure 4: Interquartile Mean (IQM) performance across 41 Atari games relative to a "No Pretrain" architecture version using diverse initialization and pretraining strategies, all maintaining consistent model size and dataset configurations.

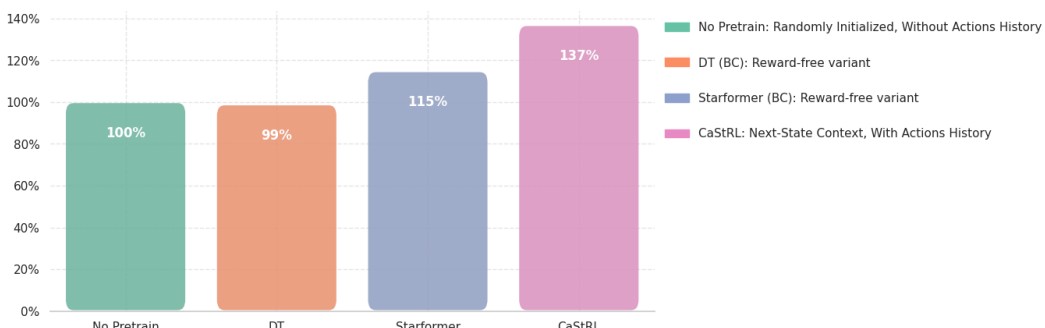

Figure 5: Interquartile Mean (IQM) performance comparison over 41 Atari games relative to "No Pretrain" architecture version of CaStRL to the baselines, with the GPT model size in DT and Starformer matched to that of CaStRL for a fair evaluation.

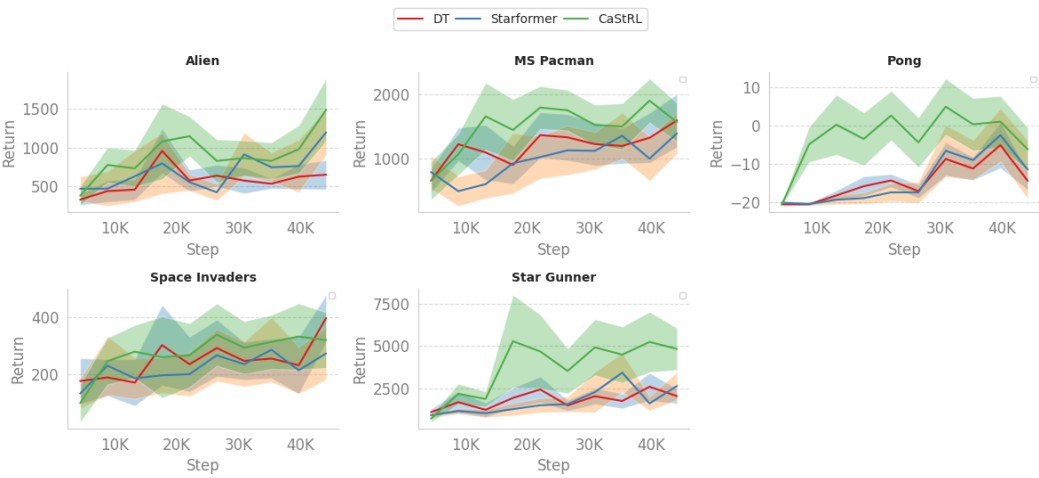

Figure 6: Fine-tuning performance on unseen environments after pretraining on a limited dataset of 41 Atari environments. The fully unsupervised pretrained CaStRL notably outperforms DT and Starformer in the ability to generate trajectories with higher returns.

The evaluation of CaStRL's scalability is based on how quickly it outperforms DT and Starformer in multi-game settings for the BC as downstream task. As shown in the Figure 6, CaStRL is much more efficient to train. Given that it's a completely unsupervised pre-training framework, its added value is expected to increase with larger-scale training, a direction we will explore in future work.

## 6    CONCLUSIONS

In this paper, we introduced Context-aware State Representation Learning (CaStRL), a novel unsupervised state representation learning approach aimed at learning generalizable state representations. Despite explicit limitations in both pretraining and finetuning, which included small dataset sizes, a limited number of epochs, and a focus on reward-free settings, CaStRL demonstrated the adaptability of the resulting state representations, making it possible to seamlessly use the learned representations across multiple environments simultaneously. Future work includes scaling up the training of CaStRL, applying CaStRL to human demonstrations for tasks even more complex than Atari, and exploring its zero-shot transfer capabilities.

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

# A    VIDEO SWIN TRANSFORMER FOR CONTROL TASKS

## A.1    TACKLING THE ABSENCE OF CAUSALITY IN VIDEO-SWIN TRANSFORMER

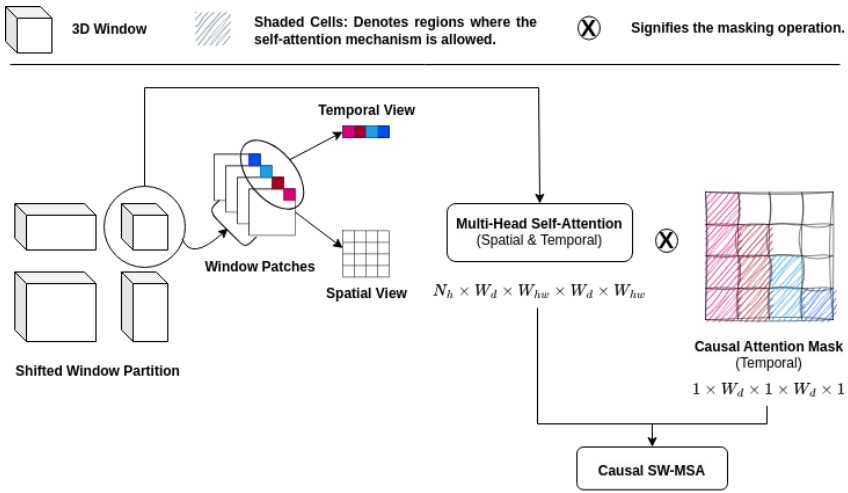

Figure 7: Illustration of Causal Shifted-Window Multi-Head self attention

In this work, we introduce a tailored attention mechanism called CSW-MSA. The incorporation of CSW-MSA seamlessly empowers the Video-Swin Transformer to excel in autoregressive control tasks, all without the need for extensive architectural adjustments. See Figure 7 for an illustration.

## A.2    INDUCTIVE BIAS BOOST: PRE-TRAINING AND SHARED KNOWLEDGE WITH CONVOLUTIONAL ENCODER

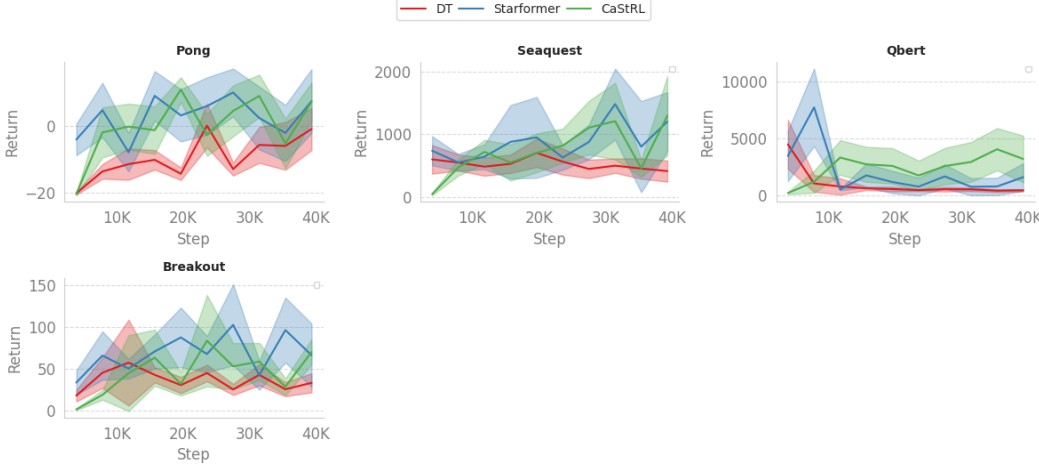

Figure 8: The finetuning performance of CaStRL in comparison to DT and Starformer.

Even though the Video-Swin Transformer (Liu et al., 2021) relies on spatiotemporal locality as an inductive bias, in control tasks, we've observed that its performance is significantly affected by initialization. This is evident when observing the performance improvement of the CaStRL model after pretraining across various Atari environments, and then finetuning the pretrained model in **single-game settings**, as illustrated in Figure 8.

---

**Algorithm 2** State Representation Learning (StRL)

---

   ▷ **T**: Sequence Length
   ▷ $\mathbf{o}^{[1:T]}$: Sequence of Observations — $o^1, o_2, \dots, o^T$
   ▷ **RandomTransformation**: Spatiotemporal Augmenter
   ▷ **ConvEncoder**: ResNet34-Based Encoder
   ▷ **TransformerEncoder**: Video-Swin Transformer with CSW-MSA
   ▷ **ConvExpander**: Expander Network for ResNet34-Based Representations
   ▷ **TransformerExpander**: Expander Network for Video-Swin Transformer Representations
1: **procedure** STRL($o^{[1:T]}$)
     % Obtaining varying viewpoints of the same sequence of observations.
2:     $\mathbf{x}^{[1:T]} \leftarrow$ **RandomTransformation**($o^{[1:T]}$)
3:     $\hat{\mathbf{x}}^{[1:T]} \leftarrow$ **RandomTransformation**($o^{[1:T]}$)
     % Encoding observational data into state representations.
4:     $\mathbf{s_x} \leftarrow$ **ConvEncoder**($x^{[1:T]}$)
5:     $\mathbf{s_{\hat{x}}} \leftarrow$ **TransformerEncoder**($\hat{x}^{[1:T]}$)
     % Extend state representations into a higher-dimensional space.
6:     $\mathbf{z} \leftarrow$ **ConvExpander**($s_x$)
7:     $\hat{\mathbf{z}} \leftarrow$ **TransformerExpander**($s_{\hat{x}}$)
     % VICReg Loss: Optimize State Representation
8:     $\mathbf{loss_{strl}} \leftarrow$ **VICReg**($z, \hat{z}$)
9: **end procedure**

---

CaStRL employs VICReg (Bardes et al., 2021) as the state representation learning objective, highlighting the necessity for two modules: a transformer module, exemplified by the Video-Swin Transformer, which possesses the capability to scale and learn spatiotemporal features, and another module, such as ResNet34, designed to compensate for the initial lack of inductive bias during the early stages of training. The pseudocode 2 illustrates the steps involved in StRL objective.

## B    CONTEXT-GPT OBJECTIVES

---

**Algorithm 3** Context GPT Loss (For Context-Aware StRL: **Context Type ← Masked State**)

---

▷ **T**: Sequence Length
▷ $\mathbf{s}^{[1:T]}$: Sequence of States — $s^1, s_2, \ldots, s^T$
▷ $\mathbf{s}^{[1:T]}_{masked}$: Sequence of Masked States
▷ **I**: Locations of Intentionally Masked Tokens.
▷ **Squeezer**: Squeezer Network for Intermediate Context Representations
▷ **ContextGPT**: GPT2-like Transformer
▷ **ContextDecoder**: Decoder Network

1: **procedure** CONTEXTGPTLOSS($s^{[1:T]}, s^{[1:T]}_{masked}, I$)
    % Project masked state into intermediate context representations.
2:    $\mathbf{c_s}^{[1:T]} \leftarrow \mathbf{Squeezer}(s^{[1:T]}_{masked})$
    % Predict the transitions context from a sequence of intermediate context representations.
3:    $\mathbf{c_{tr}}^{[1:T]} \leftarrow \mathbf{ContextGPT}(c^{[1:T]}_s)$
    % Initialize Context-GPT loss.
4:    **loss** ← 0
    % Computation of Context-GPT loss at masked locations.
5:    **for** $l$ in $I$ **do**
        % Retrieve the masked state at ($l$) using the context at ($l$).
6:        $\hat{s}^{[l]} \leftarrow \mathbf{ContextDecoders}(c^{[l]}_{tr})$
        % Accumulate the discrepancy measure between $s^{[l]}$ and $\hat{s}^{[l]}$.
7:        **loss** ← **loss** + $\mathbf{VICReg}(s^{[l]}, \hat{s}^{[l]})$
8:    **end for**
9: **end procedure**

---

**Algorithm 4** Context GPT Loss (For Context-Aware StRL: **Context Type ← Masked Action**)

---

▷ **T**: Sequence Length
▷ $\mathbf{s}^{[1:T]}$: Sequence of States — $s^1, s_2, \ldots, s^T$
▷ $\mathbf{a}^{[1:T]}$: Sequence of Actions — $a^0, a^1, \ldots, a^{T-1}$
▷ $\mathbf{E}^{[1:T]}_{a_{masked}}$: Sequence of Masked Action Tokens (Embeddings)
▷ **I**: Locations of Intentionally Masked Tokens.
▷ **Stack**: Stack $s^{[1:T]}, E^{[1:T]}_{a_{masked}}$ Sequences ← $s^1, E^0_{a_{masked}}, s^2, E^1_{a_{masked}}, \ldots, s^T, E^{T-1}_{a_{masked}}$
▷ **Squeezer**: Squeezer Network for Intermediate Context Representations
▷ **ContextGPT**: GPT2-like Transformer
▷ **ContextDecoder**: Decoder Network

1: **procedure** CONTEXTGPTLOSS($s^{[1:T]}, E^{[1:T]}_{a_{masked}}, a^{[1:T]}, I$)
    % Project masked state into intermediate context representations.
2:    $\mathbf{c_s}^{[1:T]} \leftarrow \mathbf{Squeezer}(s^{[1:T]}_{masked})$
    % Stack the intermediate context representations and action embeddings.
3:    $\mathbf{c_{sa}}^{[1:T]} \leftarrow \mathbf{Stack}(c^{[1:T]}_s, E^{[1:T]}_{a_{masked}})$
    % Predict the transitions context from a sequence of intermediate context representations.
4:    $\mathbf{c_{tr}}^{[1:T]} \leftarrow \mathbf{ContextGPT}(c^{[1:T]}_{sa})$
    % Initialize Context-GPT loss.
5:    **loss** ← 0
    % Computation of Context-GPT loss at masked locations.
6:    **for** $l$ in $I$ **do**
        % Retrieve the masked action at ($l$) using the context at ($l$).
7:        $\hat{a}^{[l]} \leftarrow \mathbf{ContextDecoders}(c^{[l]}_{tr})$
        % Accumulate the discrepancy measure between $a^{[l]}$ and $\hat{a}^{[l]}$.
8:        **loss** ← **loss** + $\mathbf{ActionPredictionLoss}(a^{[l]}, \hat{a}^{[l]})$
9:    **end for**
10: **end procedure**

---

## C    ADDITIONAL RESULTS

### C.1    CASTRL NORMALIZED SCORES

In Figures 10 and 9, we present human-normalized scores (HNS) (Toromanoff et al., 2019) and DQN-normalized scores (Agarwal et al., 2021), respectively. Normalized scores are calculated as follows:

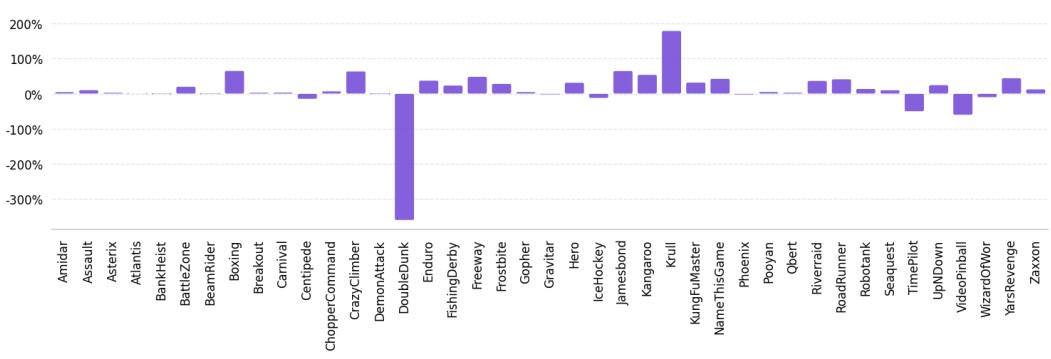

Figure 9: DQN-Normalized Scores.

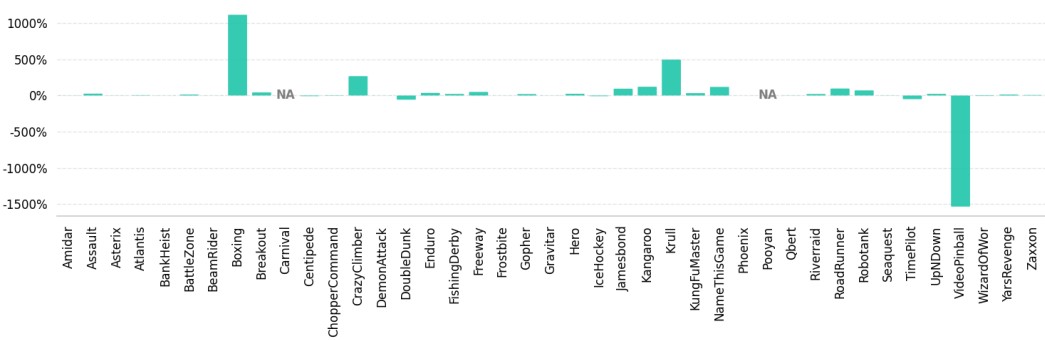

Figure 10: Human-Normalized Scores — Carnival and Pooya: **NA** (No Reference Human Scores).

$$\frac{score - score_{random}}{score_{human/dqn} - score_{random}} \tag{1}$$

## C.2 VISUALIZE ATTENTIONS

We visualize the learned representations with CaStRL using the Attention-Rollout technique (Abnar & Zuidema, 2020) to generate attention maps (as seen in Figure 11).

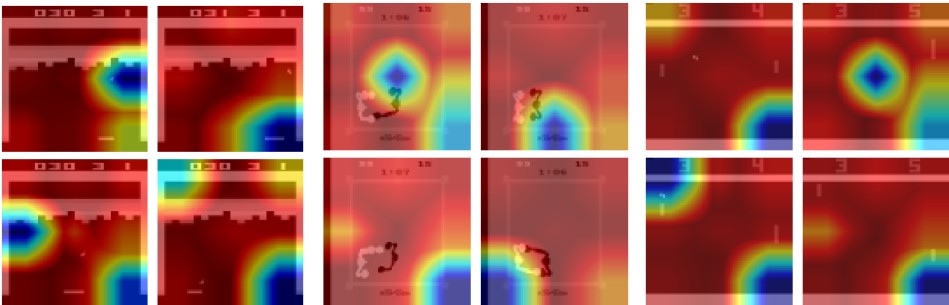

Figure 11: Visualization of attention maps in CaStRL, extracted for breakout game. We highlights the movement of ball and paddel with heat-map style coloring

# D  IMPLEMENTATION DETAILS

For both pretraining and fine-tuning, we employed the AdamW optimizer (Loshchilov & Hutter, 2017) with $\beta_1$=0.9, $\beta_2$=0.999, and $\lambda$ (weight decay) set to 0.01. We implemented a learning rate decay using a cosine annealing method described by (Loshchilov & Hutter, 2016), where $T_{max}$=2000 represents the maximum number of iterations, and the learning rate range is specified as $[\eta_{min}, \eta_{max}] \leftarrow [7\times10^{-5}, 6\times10^{-4}]$.

