# OpenReview forum: "CaStRL: Context-Aware State Representation learning with Transformer"
_ICLR.cc/2024/Conference — ICLR 2024 Conference Withdrawn Submission_

### Official Review · Reviewer_NZzK · 2023-10-20

**Soundness:** 3 good
**Presentation:** 3 good
**Contribution:** 2 fair
**Rating:** 5
**Confidence:** 4

**Summary:**

The paper introduces CASTRL, a method to generate representations from high dimensional data, useful for downstream reinforment learning applications. The authors propose to replace traditionally used vision encoder with a modified Video-Swin Transformer that learns from sequences of observations in the past. They then employ an autoregressive model to capture structure in a sequence of observations. Experimental results in reward-free settings under some computational contraints and in limited-data cases show improvements across a variety of Atari environments.

**Strengths:**

Notably a lot of current deep reinforcement learning techniques lack some structure in the latent space. This has also motivated other recent work to factorize the state space, for example by further disentangling interaction-relevant from irrelevant factor. In a similar spirit, this paper proposes a (series of) approaches on how unsupervised pretraining can help along this direction.

The authors propose a new causal masked Video Transformer and a Context GPT that learn representations based on the context of the previous observations. They propose different ways to overcome challenges along the way, as:
1. how to compensate the limited inductive bias of the Transformer under a low-data regime
2. what kind of objectives to use in order to train the state representation learning
3. what objective to use when training in a context aware way

In the end, the authors show this can be used to train a model that successfully generalizes to different environments, even in a low data  regime. The authors compare against other reward-free methods.

**Weaknesses:**

In general, I find that a lot of the proposals in the paper are not sufficiently motivated. I think it will be important for the authors to clarify these, also in the paper.

1. I find the motivation of the causal masking in the Video-Swin Transformer insufficient. It is not clear to me how much of a difference this makes, or even why this is detrimental. People have always used sequences of frames to encode the current state [1]. As far as I can tell your convolutional encoder (top part of FIgure 2) also does not use a causal mask anyway (it is not clear what "ResNet34 with minor adjustments" exactly means, unless I missed the explanation somewhere).
2. The idea of training in an unsupervised way for the state representation learning is quite nice and seems to be working. There is however a long history of related work that is related and is largely ignored in this context. Can the authors comments on what are the differences compared to [2] for example (or other object centric techniques, e.g. [3])?
3. The authors propose ContextGPT as a way to predict based on a sequence of states. The objectives proposed in Section 4.3 are interesting. I find that the results do little to motivate why the selected objective worked better in the end.

[1] Mnih, Volodymyr, et al. "Playing atari with deep reinforcement learning." arXiv preprint arXiv:1312.5602 (2013).

[2] Micheli, Vincent, Eloi Alonso, and François Fleuret. "Transformers are sample efficient world models." arXiv preprint arXiv:2209.00588 (2022).

[3] Stanić, Aleksandar, et al. "Learning to generalize with object-centric agents in the open world survival game crafter." IEEE Transactions on Games (2023).

**Questions:**

1. In the abstract you mention "significantly improves". It is not clear to me to what results exactly this is referring to.
2. Is a DQN a suitable baseline for Figure 1?
3. You use a CNN to compensate for the inductive bias of Transformers under limited data. However, it is known that with enough data Transformers become better [1]. How do you expect this tradeoff to evolve with more data?
4. Section 3.1 is interesting but is quite preliminary and disconnected from the rest of the paper. Does your method somehow guarantee better compactness, sufficiency and generalizability?
5. The y-axis in Figure 3 is not really informative. Can you perhaps change that to human normalized scores? I also do not think that you can deduce that CSW-MSA is better based on Figure 3.
6. Is there a difference between limiting the number of batch samples per epoch instead of decreasing the number of epochs? It does not make sense to me.
7. Can you comment on the model sizes that you are using? Models seems to be quite small.
8. How do you match the GPT model scale to ensure a fair comparison between CastRL and the baselines?
9. It would be interesting to provide more insights on Figure 4. For instance, why is the "Without actions history" better?
10. Is it possible to use Atari 100k Dataset for better comparison?

[1] Dosovitskiy, Alexey, et al. "An image is worth 16x16 words: Transformers for image recognition at scale." arXiv preprint arXiv:2010.11929 (2020).

---

### Official Review · Reviewer_JKQw · 2023-10-30

**Soundness:** 1 poor
**Presentation:** 1 poor
**Contribution:** 1 poor
**Rating:** 3
**Confidence:** 5

**Summary:**

The paper introduces context-aware state representation learning for reinforcement learning. The authors define 'context' as a summary of the state representation and propose a joint learning approach throughout the training process. They also present a variant of the video swin transformer that employs a causal attention mask to facilitate auto-regressive modeling. The proposed method has been tested on a broad spectrum of Atari games, where it significantly enhances downstream learning efficiency.

**Strengths:**

The exploration of high-level state representation is an intriguing direction, and I believe it holds potential value. However, the current version of the paper requires substantial improvements before it can be deemed acceptable.

**Weaknesses:**

1. The empirical results presented are unsatisfactory.
  + The performance appears to lag significantly behind the state-of-the-art. DQN-based RL methods were able to achieve a return of 20 in Pong as early as 2013 (as per 'Playing Atari with Deep Reinforcement Learning'), yet the authors' plot in Figure 6 only shows a return of ~0.
  + Figure 3, which compares models trained with and without causal attention, suggests that there is no discernible difference between the proposed method and the baseline in the games Seaquest and Pong. Furthermore, the return in these games shows negligible improvement throughout the training process, which raises questions about the effectiveness of the proposed method.
2. The paper lacks clarity and completeness, and the writing needs significant improvement.
+ Figure 2, which is a screenshot rather than a vectorized figure, is blurry and hard to interpret. The caption 'Illustration of the pre-training pipeline' does not provide sufficient clarity. The authors should make the figure more comprehensible.
+ The main focus of the paper, 'context-aware' state representation learning, is introduced quite late (on page 6), which disrupts the flow of the paper. Moreover, the paragraph describing the method lacks clarity. For instance, the term $Y'$ is not defined anywhere in the paper.
+ The pseudo code provided is confusing. For example, a 'squeezer network' is mentioned in the code block, but there is no reference to it in the main body of the paper.
+ The bar plots in Figures 4 and 5 are more suited to presentation slides than a research paper. The authors should present the raw return for each game in a tabular format for better clarity and comprehension.

**Questions:**

See weaknesses

---

### Official Review · Reviewer_6x1S · 2023-11-03

**Soundness:** 2 fair
**Presentation:** 3 good
**Contribution:** 2 fair
**Rating:** 5
**Confidence:** 4

**Summary:**

In this paper, the authors propose a Context-aware State Representation Learning (CaStRL) framework where state representation is encouraged to incorporate context information for better downstream control tasks, together with a tailored attention mechanism for the Video-Swin Transformer for sequence learning. It is shown that the proposed approach with the pretraining + fine-tuning strategy can leverage limited demonstration data to improve imitation learning for the Atari games in a cross-task transfer setup.

**Strengths:**

+ The proposed approach is simple yet effective on the shown cross-task transfer setup on 2D game environments
+ The authors have done extensive ablation studies on the pertaining strategies of their proposed method

**Weaknesses:**

- The evaluation is a bit limited. It would be better to show results with different training and held-out task splits. My current understanding is that the evaluation is performed with a fixed set of 5 held-out tasks. It will provide additional insights into how this approach benefits cross-task transfer when the task similarities are higher or lower.
- It would make the experimental evidence stronger if the authors could provide results on some other environments newer than the Atari ones (e.g., Procgen).
- Lack of comparison with other pertaining strategies for the control tasks in the experiment results (or at least in terms of discussion).

**Questions:**

- Aside from the points in "weaknesses", please clarify if the DT as the baseline is trained with the same budget as in the fine-tuning stage only of the proposed method. While I understand that it is hard to compare the baseline (w\o pertaining) fairly with the proposed method (pretraining + fine-tuning), it might be necessary to show the results of the baseline with larger fine-tuning computation budgets (e.g., train longer) to see how large the benefits are from pretraining with the proposed method.
- Missing some reference to related work. For instance, [1] also adopts the idea of context-aware representation learning for Transformer-based sequence learning in control tasks (using additional prompt tokens with a slightly different setup, though).

[1] Chain-of-thought Predictive Control